# Spatial equity in the layout of urban public sports facilities in Hangzhou

**Yujuan Chen[1], Ning Lin[1], Yangyang Wu[1], Liang Ding** [1]*, **Jun Pang[1], Tonghua Lv[2]**

**1** School of Design and Architecture, Zhejiang University of Technology, HangZhou, China, **2** Hangzhou Xuelian Land Planning Co., LTD, HangZhou, China

* liangd05813@zjut.edu.cn

## Abstract

This paper proposes a framework for a layout evaluation of urban public sports facilities. First, the buffer analysis method is used to measure the service level of public sports facilities. The study findings indicate that the overall service level of public sports facilities presents the spatial characteristics of a central agglomeration, and the value of the service level diffuses outward from high to low. There is evident spatial heterogeneity in the layout of public sports facilities in Hangzhou. Second, the Gini coefficient, Lorenz curve, and location entropy are employed to measure the equity of the distribution among spatial units and the intradistrict disparity. The results show a mismatch between the spatial distribution of the facilities and the distribution of the permanent population. The patterns of distribution of the location entropy classes of Hangzhou can be divided into three types: balanced, alternating, and divergent districts. The method in this paper is effective in measuring spatial equity and visualizing it. it has a certain degree of systemicity, universality and operability. At the same time, this method can compare the diachronic characteristics of the same city and the synchronic characteristics of different cities, which has universal application value.

**Data Availability Statement:** All data files are available from the figshare database (https://figshare.com/s/c05872042227c1a5d9fc).

## 1. Introduction

Urban space is both the physical carrier of socioeconomic development and human activities and an important determinant factor in the allocation of various resources and interests [1]. Reasonable planning of public sports facilities improves the urban space layout and facilitates the self-optimization and healthy development of an urban system [2]. As an important part of urban space, sports facilities' resources have typical social public attributes [3]. Given the significant trend in the social spatial differentiation pattern, issues related to the optimal allocation of facilities within the scope of social equity and justice have attracted significant attention from governments at all levels and all walks of life.

Equity and justice are the core values of urban planning. Influenced by the concept of spatial justice, research on the spatial equity of public service facilities has become an important research topic in the field of urban planning. Since accessibility was first proposed by Hansen [4], it has become an important indicator of the performance evaluation of equity and justice of facilities. Related content mainly focuses on the relationship between accessibility and residents' economic and social status or social needs [5–7]. For example, Omer built a spatial

**Funding:** This study was funded by the National Natural Science Foundation of China in the form of an award to YC [51808495].

**Competing interests:** The authors have read the journal's policy and have the following competing interests: Tonghua Lv became a paid employee of Hangzhou Xuelian Land Planning Co., LTD while working on this study. There are no patents, products in development or marketed products associated with this research to declare. This does not alter our adherence to PLOS ONE policies on sharing data and materials.

equity research framework for evaluating the accessibility of urban parks based on house-level census data and used the buffer method in ArcGIS [8]; Chen et al. based on the perspective of spatial equity, calculated the spatial distribution of the choice opportunities for citizens to enjoy green park space by measuring the service scope [9]. Accessibility is a commonly used tool to measure the spatial equity of public service facilities; however, most studies do not positively discuss the specific degree of equity of public service facilities' accessibility, especially the lack of comparisons of different spatial scales. Subsequently, a large number of scholars conducted research on spatial equity based on the concept of "spatial matching." Both Delbosc and Welch used the Lorentz curve and Gini coefficient methods to analyze the equity situation of public transportation resources [10, 11]. On this basis, Tang analyzed the social equity performance of rail transit and used the location entropy method to conduct a spatial visualization analysis of fairness [12]. Yang et al. introduced the social demand index and combined it with the coefficient of accessibility variation to further explain the issue of spatial equality [13].

In contrast, residents' health and sports activities are significantly correlated with the location, scale, and number of public sports facilities; therefore, the value of equity research is prominent. However, few relevant studies have been published and mainly reflect three aspects: equalization, accessibility, and optimized layout of facilities. First, the equalization research of sports facilities is divided into two dimensions: facility supply and enjoyment [14]. In terms of supply, scholars have studied the main body of the supply and governance mode of sports facilities and the spatial differences in the distribution of the facilities themselves [15, 16]. In terms of enjoyment, scholars focus on studying the use of facilities by different groups in society. Liu found through a field investigation that sports facilities in England were not proportionally matched to the population [17]. Second, in terms of accessibility, the research mainly focuses on the correlation analysis between accessibility and other factors [18]. Karen et al. used both the minimum-distance and coverage methods to investigate the relationship between playground accessibility and the population and social needs in Edmonton and to assess whether the location and quality of playground facilities are equitable [19]. Cutumisu et al. used the two-step floating catchment area method to study the association between the accessibility of sports venues and residents' physical activities [20]. Higgs et al. used the FCA model to measure the relationship between the accessibility of sports facilities and the level of regional development [21]. Finally, in the research on the layout optimization of facilities, scholars mainly solved the configuration and spatial layout of sports facilities from the perspective of urban planning management and urban policy [22–26].

Although research on the spatial equity of urban public facilities has already involved "space matching" between facilities and residents, research in this field on sports facilities is scarce, and the research methods are usually qualitative in nature, which has a certain hysteresis quality. In other words, traditional urban public sports facilities planning adopts a per capita index to attempt to ensure that the spatial allocation of public facilities reaches the goal of social equity, but it lacks an effective method to evaluate the "spatial matching" of facilities and resident population distribution. Therefore, this paper is based on the ArcGIS analysis platform and uses the Gini coefficient, Lorenz curve, and location entropy methods to construct an evaluation system to measure the spatial equity of urban public sports facilities to systemically and universally quantify and visualize the results to provide some reference for the layout planning of public sports facilities in different cities.

## 2. Methods and data

### 2.1. Research objects

Urban public sports facilities refer to a variety of venues, equipment, buildings, outdoor recreation spaces, and related services in cities (towns) that urban residents use for day-to-day

exercise and related activities to meet their health and wellness needs. The sports facilities studied in this paper do not include public open spaces, such as citizens' urban squares, parks, green spaces, and waterfronts along rivers and lakes.

## 2.2. Research methods

**2.2.1. Measurement of service levels.** The ratio between the total effective service coverage area within a spatial unit and the total area of the spatial unit is used as the quantitative indicator of the service level provided by public sports facilities allocated within the unit [27]. The formula to calculate the ratio is as follows:

$$LD_j = M_j/A_j \tag{1}$$

In the formula, *LDj* denotes the service level provided by the public sports facilities in spatial unit *j*; *Mj* is the total effective area served by all public sports facilities, namely, the total volume of public sports facilities; and *Aj* denotes the total area of spatial unit *j*.

The effective service coverage areas of public sports facilities are determined using the buffer zones, as shown in Fig 1) different classes of public sports facilities are assigned different radii for their service ranges; 2) the effective service coverage areas of various classes of public sports facilities are created through the multiple ring buffer tool in ArcGIS; and 3) different classes of sports facilities and their service coverage, indicated by concentric rings, have heterogeneous service effects; therefore, different weights are assigned to each when calculating the effective service coverage area. The following principles are considered when determining the effective service area: 1) when calculating the effective service coverage area, the areas within a spatial unit that receive services from sports facilities located outside of the spatial unit are included in the calculation of this spatial unit's effective service coverage area; and 2) if the service coverage areas of two sports facilities overlap, then the overlapping area is determined by an overlay calculation.

**2.2.2. Gini coefficient and Lorenz curve.** The Lorenz curve was first proposed by American statistician Lorenz MO in 1905 to compare the equity of wealth distribution in a region at different times or in different regions at the same time [28]. The principle is to rank the population in order of income from lowest to highest, with the cumulative percentage of population

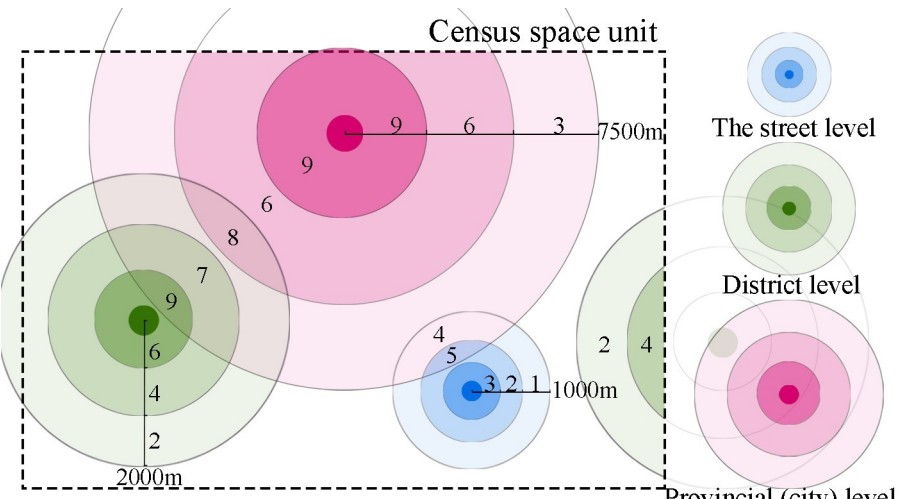

**Fig 1. Calculation diagram of effective service area of public sports facilities.**

on the X-axis and the cumulative percentage of income on the Y-axis. Based on this, the curve is drawn, and the diagonal from the origin to the end of the coordinate is the absolutely mean line. The more the curve deviates from the absolutely mean line, the more unequal the income distribution. The Gini coefficient was first proposed by Italian economist Gini C in 1912. On the basis of the Lorentz curve, the principle is to calculate the ratio of the area enclosed by the absolutely mean line and the curve to the area enclosed by the absolutely mean line and the two axes. The greater the value, the more unequal the wealth distribution [29]. In a word, the Lorentz curve is a visual representation of equality, while the Gini coefficient is a simple mathematical measure of overall inequality.

Because the essence of equity in income distribution is, to a certain extent, similar to that of equity in public resource distribution, Lorentz curve and Gini coefficient are often used in the equity research of public transportation [10, 30, 31] and green space [12], and achieved good results in exploring the equity. Therefore, public sports facilities as a part of public resources, we apply the Gini coefficient to measure equity and generate a Lorenz curve for visual presentation and analysis. First, the Gini coefficient is developed as a quantitative indicator for spatial equity in the distribution of public sports facilities, as shown in the following formula:

$$G = 1 - \sum_{k=1}^{n} (P_k - P_{k-1})(R_k + R_{k-1}) \tag{2}$$

In the formula, $P_k$ denotes the proportion of the permanent population, where $k = 1......n$, $P_0 = 0$, $P_n = 1$, and $R_k$ is the proportion of effective service coverage of public sports facilities, where $k = 1......n$, $R_0 = 0$, and $R_k = 1$. A smaller Gini coefficient results in a more equitable spatial allocation of sports facilities among all permanent residents. Typically, degrees of equity can be classified into five categories, as shown in Table 1.

Second, all of the spatial units within the study area are ranked in descending order based on public sports facilities per capita. The total permanent population is divided into intervals, with each interval containing 10% of the total permanent population; the proportion of public sports facilities utilized by each population interval is calculated and is represented by the Lorentz curve. The Lorentz curve graphically shows the distribution of public sports facilities among all permanent residents and can investigate the proportion of the permanent resident population enjoying the resources of public sports facilities. Therefore, it is an extension of the interpretation of the Gini coefficient.

**2.2.3. Location entropy.** The Location entropy was first proposed by Haggett P and applied in location analysis. Its connotation is the ratio between the proportion of an industrial sector in a certain region in the national industrial sector and the proportion of the whole industry in the region in the national industrial sector [32]. It is often used to measure the spatial distribution state of a certain production factor in a region and the degree of industrial specialization. This method is not only applied in the fields related to economy, but also gradually involved in the study of spatial equity [33, 34].

**Table 1. Gini coefficient classification table.**

| Gini coefficient value | The general meaning |
|---|---|
| < 0.2 | Absolutely equitable |
| 0.2–0.3 | Comparative equitable |
| 0.3–0.4 | Relatively reasonable |
| 0.4–0.5 | difference |
| > 0.5 | Significant difference |

The Gini coefficient and the Lorentz curve are used to measure the overall level of spatial equity, while location entropy can better display the specific spatial distribution pattern of equity. Therefore, this paper uses location entropy to analyze the spatial characteristics of spatial equity of sports facilities. The location entropy for each spatial unit is the ratio between the per capita service coverage area in the unit and the per capita service coverage area in the entire study area, as shown in formula 3:

$$LQ_j = (T_j/P_j)(T/P) \tag{3}$$

In the formula, $LQ_j$ denotes the location entropy of spatial unit $j$; $T_j$ is the effective service coverage area within spatial unit $j$; $P_j$ is the total population within spatial unit $j$; $T$ is the effective service coverage area within the study area; and $P$ is the total population of the study area. If a spatial unit has a location entropy greater than 1, then the unit's per capita service provided by public sports facilities is higher than that of the study area, and vice versa.

## 2.3. Data sources

The study region is Hangzhou, China, which includes six districts with relatively well developed and maintained urban infrastructure: Shangcheng, Xiacheng, Gongshu, Xihu, Jianggan, and Binjiang. Based on 2017 statistics, the study region has a total land area of 707.59 kilometers and a total population of 4,412,855. The region contains 49 urban subdistricts (towns) and 637 communities. The geographic center of the population distribution in each spatial unit is used as the center to divide the city into concentric rings. The area within five kilometers of the center is the central ring; the area within 5 to 10 kilometers is the transition ring; and the area beyond 10 kilometers is the suburban ring.

The 2017 cross-sectional data of population are from the publicly available government data and the open Internet platform. Because China's population census is conducted every ten years, the sixth census (2010) data is too old and the seventh census (2020) data has not yet been released. In 2017, Hangzhou municipal government conducted a miniature population By-census. The data is made public by the government (https://data.hz.zjzwfw.gov.cn/). A census of Hangzhou which is a large city with a population of nearly ten million is not conducted every year. Therefore, based on the minimum research unit in this paper, the 2017 data are the most recent available before the 7th census is released. However, the 2017 public data of the government were only counted at the street level. To obtain population data at the community level, this paper relied on the heat map of the Baidu Map Open Platform. Combined with the existing population data at the street level, the population of the community unit is determined according to the proportion of the area of the heat grid in the community unit to the total area of the street in which the community is located; in this paper, the map image of Hangzhou comes from the National Platform for Common Geospatial Information Services (https://www.tianditu.gov.cn/), a public welfare service website and does not involve copyright.

The data on public sports facilities are extracted from the 2017 point of interest (POI) data in AutoNavi Maps, 2017 aerial images in Google Earth, and the 2017 Statistics for Sports Venues and Facilities in Zheiang Province. Although urban public sports facilities have access to the latest data, their 2017 data are still used to be consistent with the population data. The main purpose of this paper is to establish an evaluation framework. Therefore, although there is a gap of 3 years between the data used in 2017 and reality, the data in this paper should still be applicable to method construction.

"Sports and leisure facilities" are selected from the above POIs and then compared with satellite images. Invalid and repetitive POIs are eliminated, and the facilities of colleges and universities, polytechnic schools, and high schools are added. As a result, a database of 181 data

items is developed. Furthermore, the facilities are classified into three categories: provincial (city) level (n = 7), district level (n = 13), and subdistrict level (n = 161).

## 2.4. Evaluation model

A model is developed to study the spatial equity in the distribution of sports facilities, as shown in Fig 2. In the first step of the evaluation, concentric rings are created to measure the service scopes of public sports facilities at different levels. Then, the service level provided by the public sports facilities in each spatial unit is measured, a chart is created to show public sports facilities' spatial distribution of service levels, and the overall service level is calculated by overlaying the services provided by all levels of sports facilities. An analysis of the service level differences among the concentric rings is performed. The second step involves the following procedures: the Gini coefficient is applied to intuitively present the distribution of public sports facilities among permanent residents; a Lorenz curve is developed to visually demonstrate the spatial equity; a location entropy is employed to measure equity in the distribution among spatial units; and the spatial units are classified into five service levels based on the location entropy—"extremely low, low, medium, high, and extremely high"—to analyze the service level differences between the concentric rings and the intradistrict disparity.

Sports facilities' service scopes are an important indicator for measuring accessibility to and disparities in the service level of these facilities. Residents who choose different public facilities also choose different modes for moving around a city (walking, bicycling, and taking public transit), and the time and distance that are acceptable for residents regarding moving around the city also differ. Usually, when choosing a higher-level facility, residents tend to accept higher time and distance thresholds. The time and distance involved with different modes of

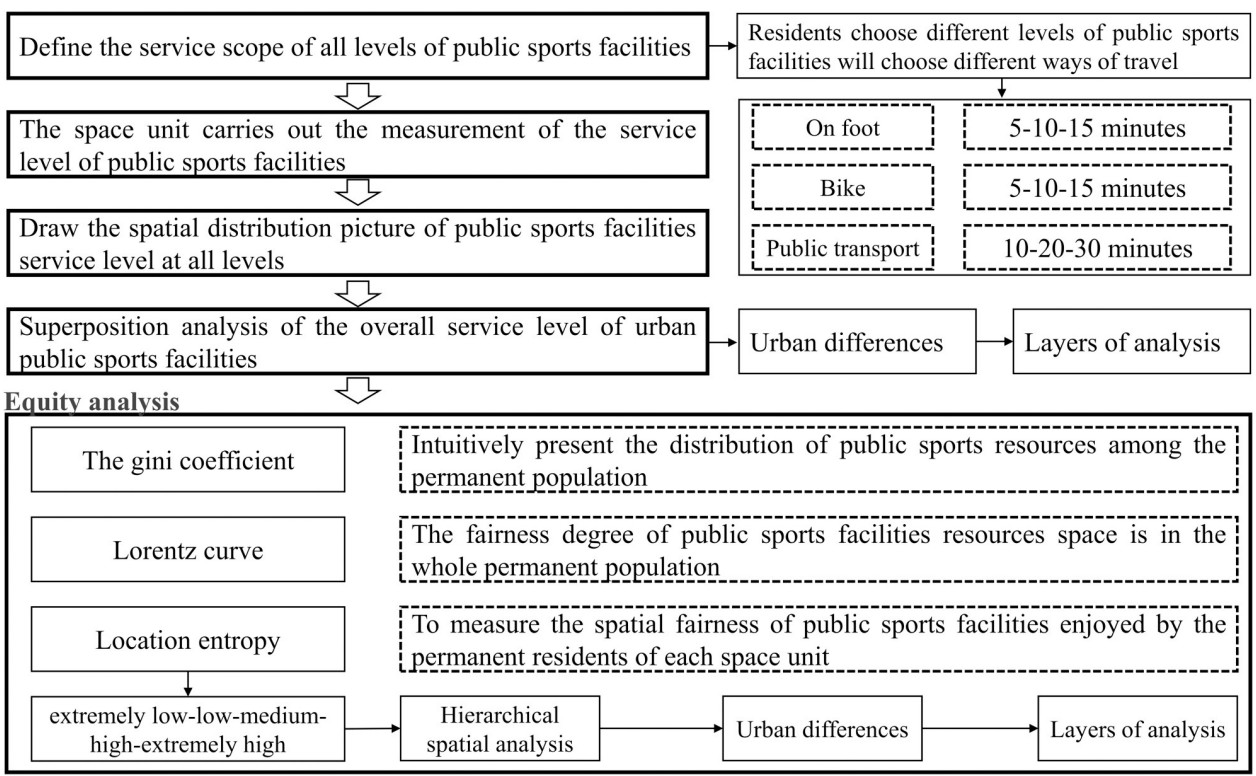

**Fig 2. Research model.**

**Table 2. Time and distance corresponding to different modes for moving.**

| Modes for moving | Time/minute | Distance/meter |
|---|---|---|
| Walking 1M/s | 5 | 300 |
| | 10 | 600 |
| | 15 | 900 |
| Bicycling 2.3 M/s | 5 | 700 |
| | 10 | 1400 |
| | 15 | 2100 |
| Taking public transit 15 Km/h | 10 | 2500 |
| | 20 | 5000 |
| | 30 | 7500 |

transportation are identified (Table 2) with assistance from the normative requirements for sports facilities in the national standards that is Standard for urban public service facilities planning (GB50442-2015) and different travel characteristics [35].

Distance is a criterion for measuring the service capacity of public sports facilities. The closer residents are to a sports facility, the more they benefit from the facility. Therefore, based on the comprehensive consideration of facility levels and their service coverage range, scores are assigned to the three concentric rings for various service levels provided by public sports facilities, as shown in Table 3. A higher-level facility has a higher score, and a shorter distance to a facility also results in a higher score.

## 3. Results

### 3.1. Service level analysis

**3.1.1. Spatial distribution of the service level of public sports facilities.** The service level of sports facilities in each spatial unit is obtained through measurements. A chart showing the spatial distribution of different levels of sports facilities in Hangzhou was created using ArcGIS software (Fig 3). Based on the statistics and spatial analysis, the following conclusions are drawn.

1. As indicated in Fig 3A, provincial (city)-level public sports facilities only account for 3.87% of all sports facilities in Hangzhou; however, given the large service range, their coverage is very wide. Based on a service range of 7,500 meters, this class of facilities covers an area of 515.53 km$^2$, that is, 72.89% of the study area. The service level of provincial (city)-level facilities is high in the city center and low on the east and west sides. Service-level peaks are

**Table 3. Service radius and evaluation standard of public sports facilities at all levels.**

| Facility level | Service radius (meter) | Evaluation criterion |
|---|---|---|
| The street level | < 300 | 3 |
| | 300–600 | 2 |
| | 600–900 | 1 |
| District level | < 700 | 6 |
| | 700–1400 | 4 |
| | 1400–2100 | 2 |
| Province city level | < 2500 | 9 |
| | 2500–5000 | 6 |
| | 5000–7500 | 3 |

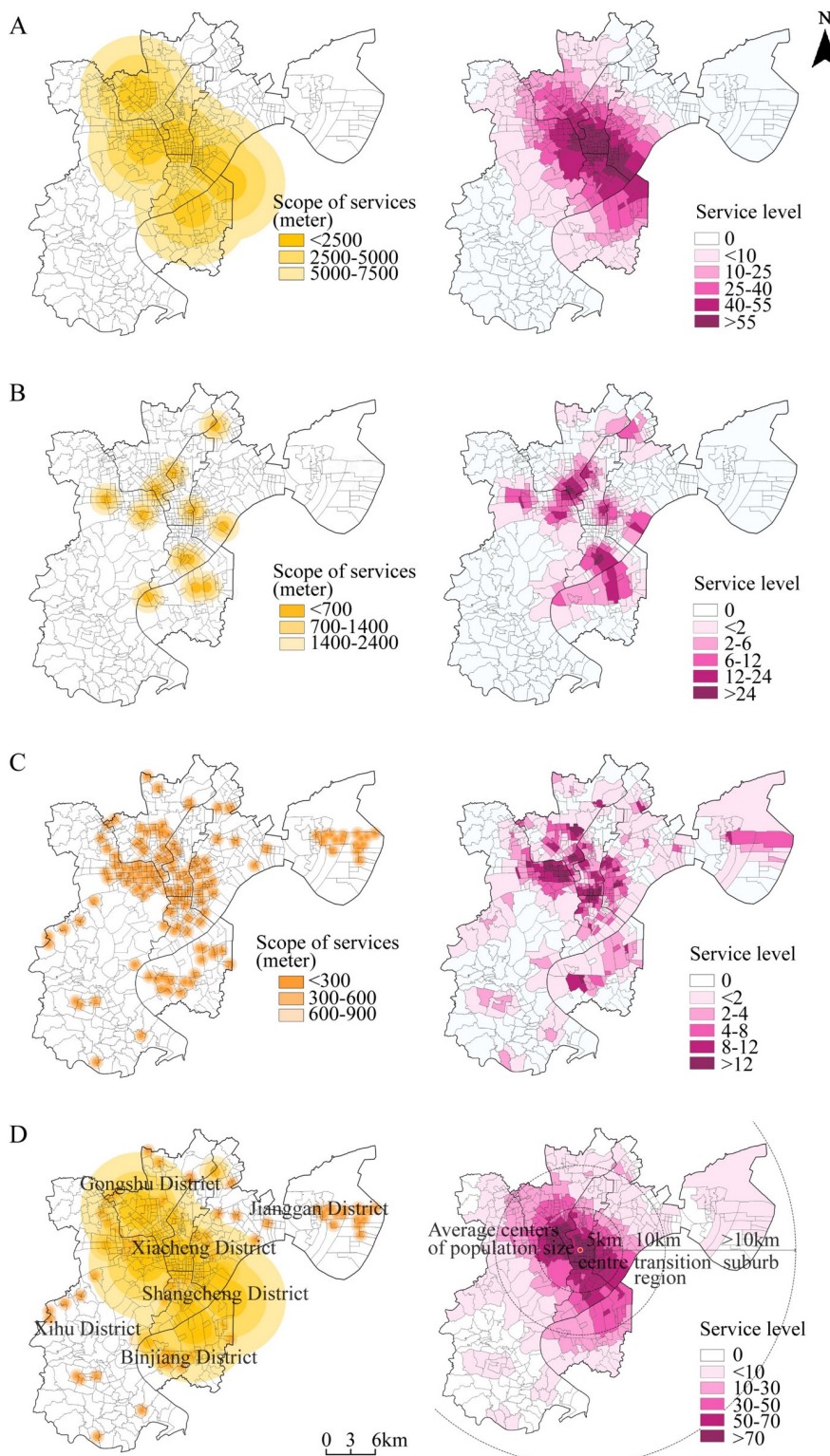

**Fig 3.** Spatial distribution of effective service coverage range and service level of sports facilities in Hangzhou: (A) Spatial distribution of effective service scope and service level of public sports facilities at province(city)-level; (B) Spatial distribution of effective service scope and service level of public sports facilities at District-level; (C) Spatial distribution of effective service scope and service level of public sports facilities at Subdistrict-level; (D) Spatial distribution and rings distribution of overall effective service scope and service level of public sports facilities. Source:

Created by the author based on the base map of Hangzhou which comes from the National Platform for Common Geospatial Information Services (https://www.tianditu.gov.cn/).

mainly located at lakefront areas along the border of Xiacheng District and Shangcheng District, in the portion of Wulin District that borders Xihu District, and along both sides of Tianmushan Road and the elevated portion of Zhonghe Road, forming a cross-shaped peak service zone. This configuration indicates that the service level provided by provincial (city)-level facilities clusters around city centers; residents living within these zones can fully enjoy these facilities, but residents outside of the zones benefit little from them. The major reason for this disparity is that the construction of large-scale sports facilities lags behind urban development. From the West Lake area to the Qiantang River area, from cross-river development to embracing river development, Hangzhou has seen continuous growth in its urban spaces; however, the development of large-scale public sports facilities has fallen short.

2. As indicated in Fig 3B, district-level sports facilities are evenly distributed in the central part of the city, with a shortage on both the east and west sides. The number of sports facilities at this level is small, accounting for 7.18% of all facilities in Hangzhou. Based on a service range of 2,100 meters, these facilities cover an area of 114.52 km$^2$, accounting for only 20.43% of the study area. The service level distribution has two features: 1) overall, service levels form discrete patches, and the Wulin business zone has an extremely low level, and 2) high-service-level zones deviate from the geometric centers of the districts. The goals of district-level facilities are to meet the sports activity needs of residents within each district and to host district-level cultural, sports, and wellness activities. Hangzhou performs well regarding developing district-level sports facilities and can meet the needs of the majority of residents in their respective districts. Given the limit of service scopes and rings, there are some "service blind zones."

3. Fig 3C indicates that subdistrict-level facilities cover large areas west and north of West Lake and that service is relatively concentrated in Xiasha District and Binjiang District; this is consistent with the planned urban layout (the revised Master Plan of 2016) that envisions "a major city with three subcities.". There are a large number of facilities at this level, accounting for 88.95% of all of the facilities in the city. Based on a 900-meter service radius, the actual service coverage area is 233.12 km$^2$, that is, 32.95% of the study area. Overall, the service level of these sports facilities is broad in the central parts and spotty in the periphery of the city centers. Peak service levels are observed in three subdistricts: Cuiyuan, Shangtang, and Xiaoying. Isolated spatial units with high service levels can be found in the subdistricts of Banshan, Dinglan, Puyan, and Baiyang. Different from provincial (city)-level and district-level facilities, subdistrict-level facilities provide a high service level in the subdistricts of Liuxia, Zhuantang, Pengbu, and Jiubao. Subdistrict-level facilities aim to meet residents' sports activity needs, such as ball games and swimming; the goal is to provide sports facilities within a 15-minute commute. Facilities in this class require relatively low construction costs, cover a greater total area than do district-level facilities, and serve a larger number of subdistricts than do provincial (city)-level facilities.

From needs to actual selection, residents' choice of sports activities involves a complex process, and time, space, and the variety of activities influence their choice. Facilities that offer different service levels supplement each other. To perform a comprehensive evaluation of the service level of all public sports facilities in the urban areas of Hangzhou, the service coverage

of all three levels of facilities is overlaid, generating a spatial distribution of the consolidated service coverage and service levels. As shown in Fig 3D, only Shangcheng District is fully serviced by public sports facilities; to varying degrees, the other five districts have "vacuum zones" of coverage. Xihu District (the towns of Zhuantang and Shuangpu, and the Sandun and Xihu subdistricts) and Jianggan District (the subdistricts of Xiasha and Jiubao) have many zones that are outside the effective coverage area—an indication that people in the city center areas of Hangzhou have access to well-developed sports facilities. However, the urban villages, factories, villages, and farmland on the outskirts of city centers are still awaiting urban development. As a result, there is no effective service coverage of sports facilities in these areas. Overall, areas with combined service levels are clustered around city centers, resulting in two interconnected smaller centers around the Xixi and Changqing-Chaoming subdistricts. The service level decreases with increasing distance from the center. The disparity in service levels results from the resource advantages of city centers and the lag in development in the periphery of cities. However, given the impact of mountains and water bodies and the differential land rent theory, service level and distance to the city center do not constitute a simple linear relationship.

**3.1.2. Service level disparity among districts.** Based on formula 1, each district's service level is derived. Based on the statistics for the service level provided in the spatial units within each district, the average and median service level is calculated for each district. Furthermore, a chart showing the service levels in all districts is created, as shown in Fig 4. The overall service level provided by all of the sports facilities within the study area has the following evident spatial disparities.

1. In terms of the concentric rings, the service level in the center ring is significantly higher than that in the outer rings; the transition ring also shows extremely high disparity.

2. In terms of service level by district, the service levels provided by sports facilities in Shangcheng District and Xiacheng District are significantly higher than those in other districts; these districts constitute the upper echelon, followed by Binjiang District and Gongshu District. Jianggan District and Xihu District have the lowest service levels.

3. In terms of the average service level in the spatial units within each district, Shangcheng District and Xiacheng District have equivalent averages, and both belong to the upper echelon. However, only Gongshu District remains in the second echelon; the average service

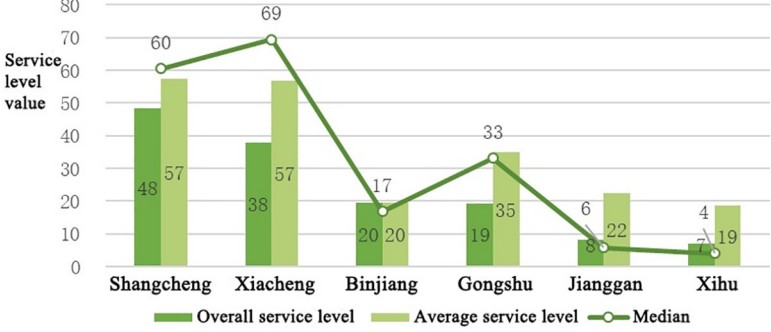

**Fig 4. Service level disparity of public sports among each district.** Overall service level (darker green column) represents the sum of each subdistrict service level in each district; Average service level (lighter green column) represents the average of the sum of the service levels of each subdistrict in each district; The line represents the median of service level in each district.

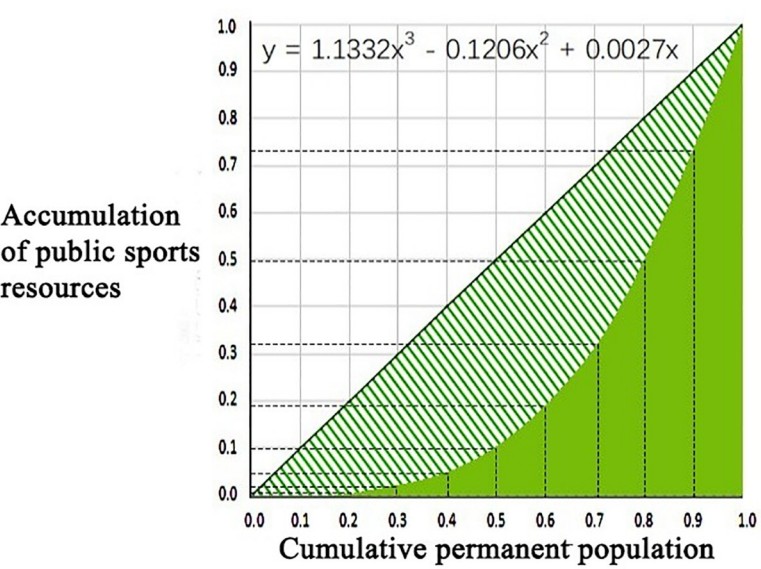

$$y = 1.1332x^3 - 0.1206x^2 + 0.0027x$$

**Fig 5. Lorenz curve of resource allocation of public sports facilities.**

level in Binjiang District is significantly lower than that in Gongshu; thus, the service levels in Binjiang District, Jianggan District, and Xihu District are in the third echelon.

4. In terms of the median service level in the spatial units in each district, the median in Xia-cheng District is higher than that in Shangcheng District, and both medians are higher than their respective averages. These districts are closely followed by Gongshu District and Bin-jiang District, which have medians that are close to their respective averages; the medians in Jianggan District and Xihu District are significantly lower than their respective averages.

## 3.2. Evaluation of equity

**3.2.1. Analysis of the Gini coefficient and Lorenz curve.** The Gini coefficient is used to evaluate equity in the distribution of public sports facilities. Through formula 2, the 2017 Gini coefficient is estimated to be 0.51. Based on Table 1, it can be concluded that the distribution of public sports resources in the study area is extremely uneven. This unevenness, to a certain extent, indicates that the construction of public sports facilities lags behind urban expansion. To further demonstrate the unbalanced distribution, a Lorenz curve that depicts the allocation of public sports resources among the permanent population is created (Fig 5); this chart indi-cates extremely high and low concentrations in the allocation of sports resources among the permanent population. Table 4 provides more information about the inequitable distribution of sports facilities: 10% of permanent residents do not have access to any public sports resources; 30% of permanent residents have access to only 2% of public sports resources; 10% of permanent residents have access to 27% of sports resources; and 30% of permanent resi-dents have access to 66.9% of resources.

**Table 4. Cumulative list of the proportion of permanent residents having access to public sports facilities resources.**

| The cumulative proportion of permanent residents (%) | 10 | 20 | 30 | 40 | 50 | 60 | 70 | 80 | 90 | 100 |
|---|---|---|---|---|---|---|---|---|---|---|
| The cumulative proportion of sports facilities resources (%) | 0.0 | 0.3 | 1.9 | 5.3 | 11.2 | 20.2 | 33.1 | 50.4 | 73.0 | 100 |

**Table 5. The number and proportion of spatial units of location entropy.**

| Level | Location entropy | Number of units | Proportion | Unit area (km$^2$) | Proportion |
|---|---|---|---|---|---|
| Extremely low | < 0.5 | 249 | 39.09% | 398.3 | 56.29% |
| Low | 0.5–0.8 | 110 | 17.27% | 56.8 | 8.03% |
| Medium | 0.8–1.2 | 87 | 13.66% | 43.1 | 6.10% |
| High | 1.2–2.0 | 76 | 10.52% | 56.0 | 7.91% |
| Extremely high | > 2.0 | 115 | 18.05% | 153.5 | 21.70% |

**3.2.2. Location entropy.** The location entropy is an indicator for measuring the factor distribution in a region and reflects the degree of specialization of an industry. This study introduces the location entropy into the analysis of spatial equity in the distribution of sports facilities. If the location entropy of a spatial unit is greater than 1, then the sports facilities per capita in the unit is higher than that in the study area, and vice versa. The location entropy for the spatial units is derived from formula 3. To make the data more intuitive and to better inform development strategies, the spatial units are classified into five service levels based on the location entropy, as shown in Table 5. A spatial distribution chart is also created based on this information, as shown in Fig 6.

1. There are more spatial units in the two extreme classes than in the classes in the middle. There are 249 spatial units in the extremely low class, followed by 115 spatial units in the extremely high class. There are fewer units in the low, medium, and high classes relative to those in the above two classes.

2. Spatial units with a location entropy less than 0.8 account for a large proportion (56.36%) of the total number of units, and their area accounts for an even larger proportion (64.32%) of the total study area. Spatial units with a location entropy greater than 1.2 account for a small proportion—less than 30% in terms of the number of units and the total area.

3. The periphery of cities in the study area mainly contains spatial units with low and extremely low location entropy, although there are a few isolated units with high location entropy. The spatial units with high or extremely high location entropy display a continuous "surrounding" distribution pattern; the spatial units with location entropy between 0.5 and 1.2 are distributed in clusters and "surrounded" by the abovementioned area. Furthermore, the spatial units within the "cluster groups" with medium location entropy outnumber the "cluster groups" with lower location entropy.

4. From the perspective of concentric rings, spatial units in the center ring have low location entropy; spatial units at the interface between the center and transition rings have high location entropy; and the spatial units in the suburban ring have the lowest location entropy.

**3.2.3. Spatial disparity by location entropy class.** The spatial units are further analyzed by location entropy class, as shown in Fig 7:

1. Areas with extremely low location entropy (less than 0.5)–These spatial units are concentrated in large patches at the periphery of urban areas and include the subdistricts of Zhuantang, Sandun, Xiasha, Jiubao, Jianqiao, Shiqiao, and Banshan. These low-entropy spatial units account for 39.09% of the total number of spatial units, and their area accounts for 56.29% of the study area. There is an extreme shortage of sports facilities in these areas, and they should be priority areas in future sports facility planning.

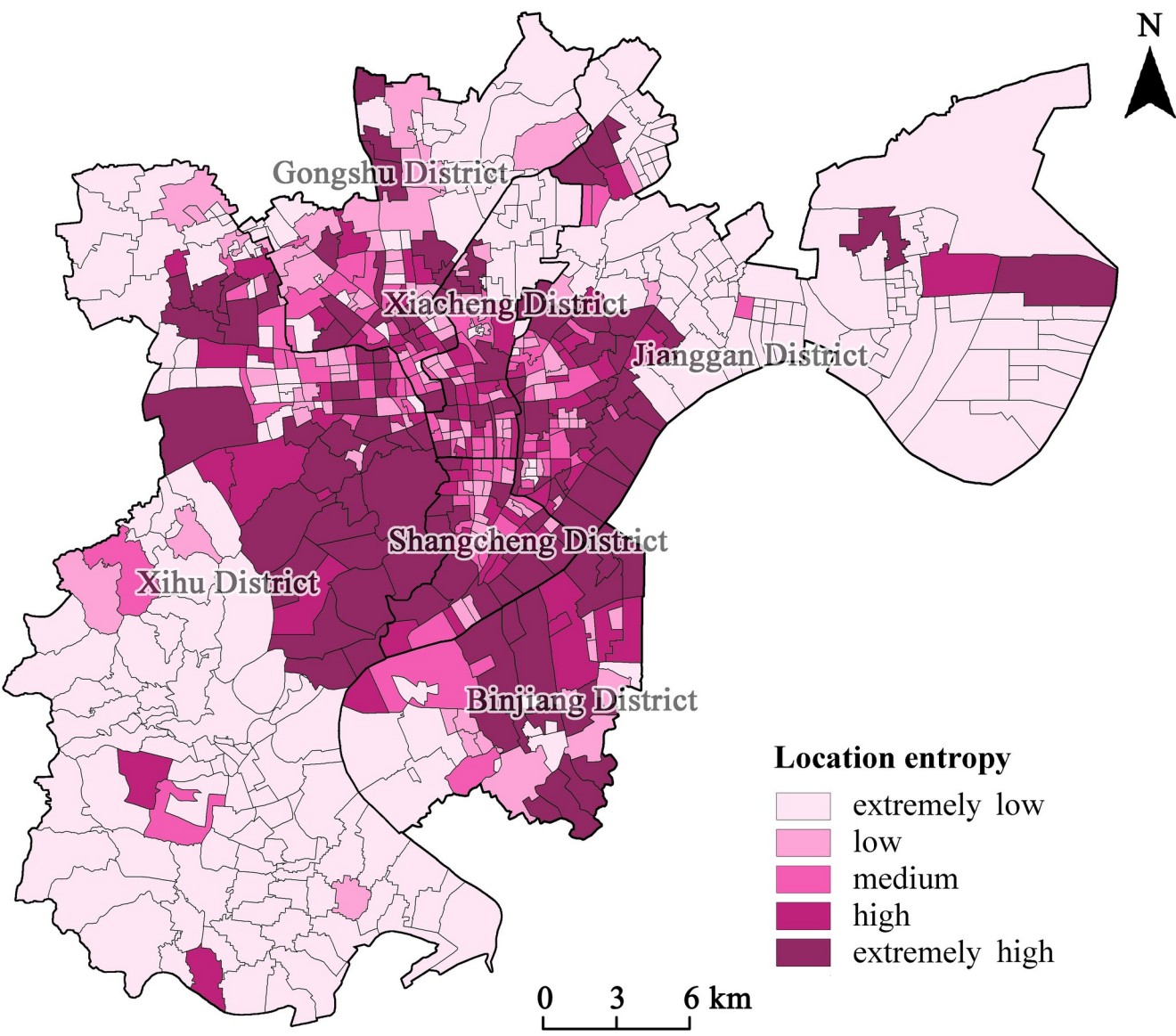

**Fig 6. Spatial distribution pattern of per capita public sports facilities resources based on location entropy allocation.**

2. Areas with low location entropy (0.5–0.8)–These spatial units are mainly concentrated in large patches at the periphery of urban areas. More specifically, some of these low-entropy units are located in patches on the north side of the city in subdistricts including Kangqiao, Shangtang, and Xiangfu; the other spatial units in this class are mainly clustered in city centers. The spatial units in this class account for 17.27% of the total number of spatial units, and their area accounts for 8.03% of the study area. Overall, there is a scarcity of sports facilities in these areas, and they do not meet the exercising needs of people in high-population-density areas.

3. Areas with medium location entropy (0.8–1.2)–These spatial units show certain concentrations at the borders between districts, such as the border between Shangcheng District and Xiacheng District and the border between Gongshu District and Xihu District. They account for 13.66% of the total number of spatial units, and their area accounts for 8.03% of

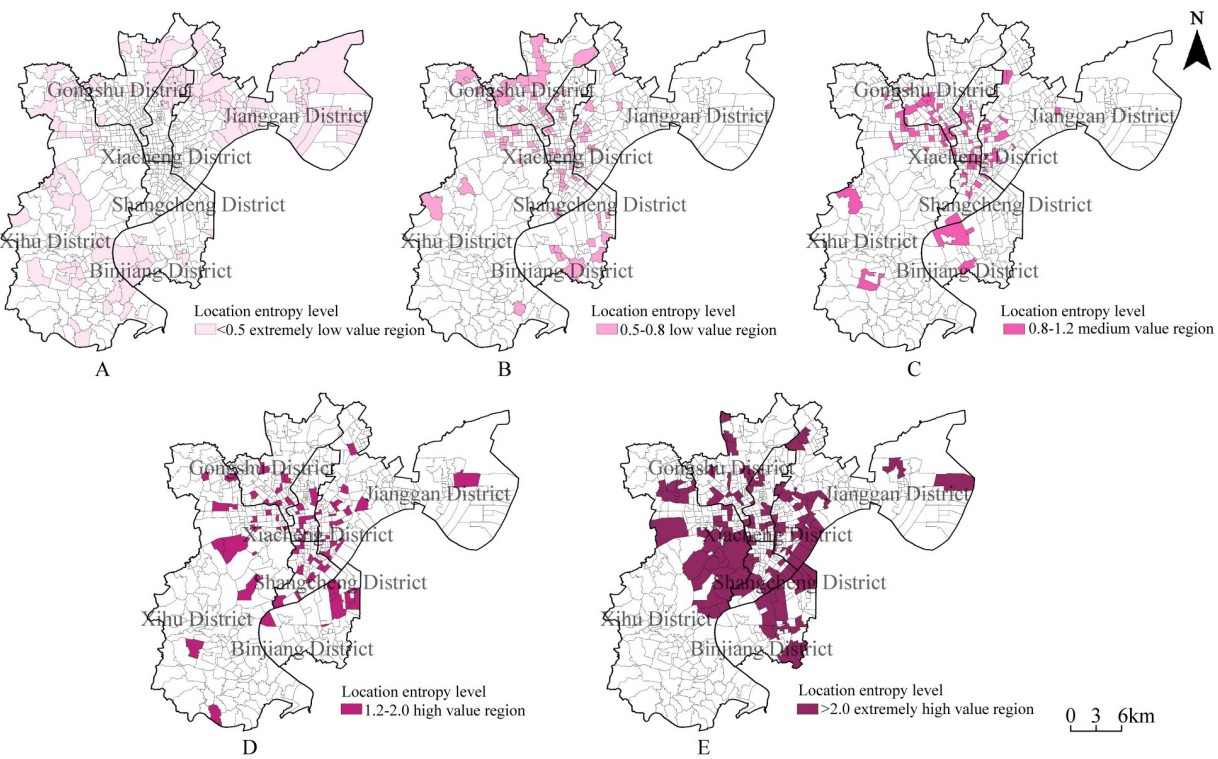

**Fig 7. Regional distribution map of each level of location entropy.** (A) Areas with extremely low location entropy (less than 0.5); (B) Areas with low location entropy (0.5–0.8); (C) Areas with medium location entropy (0.8–1.2); (D) Areas with high location entropy (1.2–2.0); (E) Areas with extremely high location entropy (greater than 2.0). Source: Created by the author based on the base map of Hangzhou which comes from the National Platform for Common Geospatial Information Services (https://www.tianditu.gov.cn/).

the study area. These areas represent the average level of sports facility development in Hangzhou.

4. Areas with high location entropy (1.2–2.0)–These spatial units show certain concentrations at the borders between districts. This class includes spatial units in transition between the medium and extremely high classes. Spatial units in this class account for 10.52% of the total number of spatial units, and their area accounts for 7.91% of the study area.

5. Areas with extremely high location entropy (greater than 2.0)–These spatial units are mainly located close to natural ecological sites, such as West Lake, Xixi Wetlands, the Qiantang River, and the Grand Canal. They account for 18.05% of the total number of spatial units, and their area accounts for 21.70% of the study area.

**3.2.4. Intradistrict disparity.** A district-level review is required to evaluate spatial equity in the distribution of public sports facilities in Hangzhou. This section mainly examines the intradistrict disparity in the distribution based on location entropy.

The patterns of distribution of the location entropy classes within the six districts can be divided into three types: balanced (Shangcheng District), alternating (Xiacheng District, Gongshu District, and Binjiang District), and divergent (Xihu District and Jianggan District) districts. As shown in Fig 8, in the balanced districts, each subdistrict has spatial units that belong to high- or low-location-entropy classes, and the overall distribution is balanced. In the alternating districts, the location entropy classes of the spatial units take on an obvious

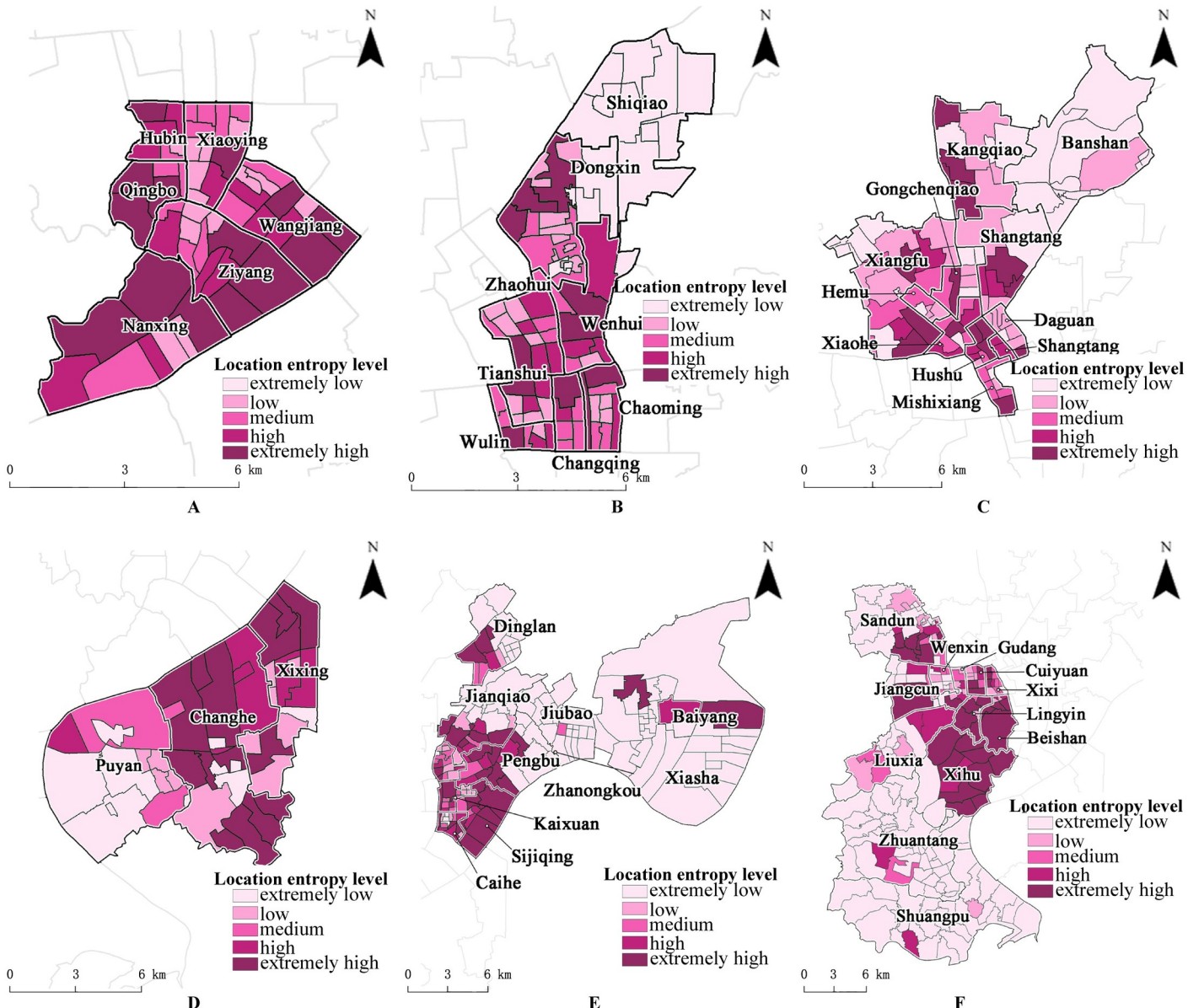

**Fig 8. Distribution pattern of location entropy in each district.** (A) Distribution pattern of location entropy in Shangcheng District; (B) Distribution pattern of location entropy in Xiacheng district; (C) Distribution pattern of location entropy in Gongshu District; (D) Distribution pattern of location entropy in Bingjing District; (E) Distribution pattern of location entropy in Jianggan District; (f) Distribution pattern of location entropy in Xihu District. Source: Created by the author based on the base map of Hangzhou which comes from the National Platform for Common Geospatial Information Services (https://www.tianditu.gov.cn/).

alternating pattern. In the divergent districts, the location entropy classes of the spatial units display an evident pattern of extremes, with a large number of extremely high-class/extremely low-class spatial units.

## 4. Discussion

### 4.1. Contributions to research analysis methods

This paper provides a complete method system for the evaluation and analysis of the spatial equity of the distribution of public sports facilities. The method has a certain degree of

systemicity, universality, and operability. On the one hand, traditional urban public sports facilities' planning used per capita indicators and service radius to attempt to ensure that the distribution of sports facilities meets the goal of social equity. However, it is based on the homogeneity of urban social space. There is a lack of effective methods to evaluate the performance of the "spatial match" between the distribution of public sports facilities and permanent populations. Therefore, this paper focuses on the three factors of distance, scale, and population density, starting from the perspective of equity in opportunities, and constructs a content system to analyze the equality of the distribution of public sports facilities from the perspective of space matching. On the other hand, this paper adopts the Gini coefficient and the Lorenz curve to evaluate the overall level of the "spatial match." This method can compare the diachronic characteristics of the same city and the synchronicity of different cities and has universal application value. In addition, the location entropy method is used to further investigate the spatial pattern of facility distribution equality, and the research results are further visualized and presented in a refined manner, making the analysis and discussion more intuitive and effective. This study has certain reference significance for various service facilities.

## 4.2. Contributions to the optimization of the layout of public sports facilities

At present, given differences in economic income, cultural value orientation, age structure, and other aspects, large cities are experiencing a dual process of social class differentiation and social space polarization, which has led to an actual contradiction in the mismatch between the supply and demand of public sports facilities. The vast majority of urban public sports facilities are occupied by a very small number of permanent residents, which is caused by the concentration of facilities and population in the city center. This research can provide the following assistance for the optimization of the distribution of public sports facilities. On the one hand, it can effectively identify areas with low spatial equity performance, promote the facility layout to incline to areas with low value, and continuously strengthen the planning and construction of public sports facilities in such areas to achieve the "spatial match" between the supply and demand of facilities. On the other hand, in the layout of public sports facilities, more attention needs to be paid to the needs of "people." In the past, research on the layout of facilities mostly considered the differences between different regions and urban and rural areas and ignored the characteristics of the needs of different groups in the city. Therefore, a foundation is laid for research on the layout of public sports facilities based on the needs of disadvantaged groups.

## 4.3. Limitations and future research

There are still many points worth discussing in studies on the spatial equality of the distribution of public sports facilities, which is also the focus of future research. First, regarding the measurement of the service level of public sports facilities, limited by the availability of data, this paper adopts the buffer analysis method, which has a certain gap with the actual situation. Although the network analysis method does not have a fundamental impact on the research results, if it can be used on the basis of obtaining the corresponding data, the research results can be more accurate. Second, the fixedness of facilities and the mobility of the service population is a difficult point in the spatial match. The cross-sectional data of the population used in this study are major limitations. In the future, mobile Internet big data applications, including the real-time distribution of the residential population, the employed population, and the recreational population, should be used to provide diversified data sources. Finally, this paper only conducts empirical research based on the data of a specific city in a certain year. A

diachronic comparison of the same city and a synchronic comparison between different cities can be carried out in the future.

## 5. Conclusion

This paper constructs a complete research framework for evaluating the spatial equity of facilities, which is mainly divided into three levels. First, the quantitative index of sports facilities resource level is obtained by using buffer analysis and superposition analysis. Secondly, Gini coefficient and Lorenz curve are used to reflect the equity of public sports facilities resources in the spatial distribution of the whole resident population. Finally, in order to present the results more finely, the spatial pattern of equity of public sports facilities was analyzed by using location entropy method.

This paper takes Hangzhou as an example to carry out empirical research and finds that the effective service scope of the central urban area of public sports facilities in Hangzhou is relatively complete, but a "vacuum" area in the periphery still persists. The service level generally presents the spatial characteristics of agglomeration with Xixi Street and Changqing-Chaoming Street as the core, and the level value diffuses outward from high to low. The central area is obviously higher than other circles, and the transition area also presents obvious differentiation of two levels. Second, it is found that the distribution of public sports facilities resources in Hangzhou in 2017 is unbalanced because of the construction level of sports facilities in Hangzhou behind the pace of urban construction expansion. The distribution of public sports resources in the permanent population shows the characteristic of "more in the middle and less at the ends." Finally, in general, the regions with high location entropy value are concentrated in the central region, whereas the outer region is dominated by spatial units with very low location entropy and relatively low level. In contrast, the location entropy level distribution pattern of six urban spatial units can be roughly divided into three types: balanced urban area(Shangcheng District), alternating urban area (Xiacheng District, Gongshu District, and Binjiang District), and divergent urban area (Xihu District and Jianggan District).

Spatial heterogeneity in the distribution of public sports facilities in Hangzhou causes spatial division, monopolizes residents' sports activities, and limits the performance of public sports facilities. Different areas are entitled to different service levels provided by various sports facilities, and an absence of these options constitutes spatial inequity. This study can be expanded by collecting historical data that reflect the dynamic changes in the distribution of sports facilities over time. Based on the data, a longitudinal analysis of the development and spatial layout of sports facilities during different periods can be performed. This historical perspective will further inform decision making regarding the provision of sports facilities in urban areas.

## Author Contributions

**Conceptualization:** Yujuan Chen.

**Data curation:** Yangyang Wu, Tonghua Lv.

**Formal analysis:** Yangyang Wu.

**Investigation:** Jun Pang.

**Methodology:** Yangyang Wu.

**Project administration:** Yujuan Chen.

**Resources:** Yujuan Chen, Liang Ding.

**Software:** Liang Ding.

**Supervision:** Liang Ding.

**Validation:** Yujuan Chen, Liang Ding.

**Visualization:** Ning Lin.

**Writing – original draft:** Yujuan Chen, Yangyang Wu.

**Writing – review & editing:** Ning Lin, Liang Ding.

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
