## [Decision Letter · Decision Letter 0]

27 Apr 2021

PONE-D-21-10518

Spatial Equity in the Layout of Urban Public Sports Facilities

PLOS ONE

Dear Dr. Ding,

Thank you for submitting your manuscript to PLOS ONE. After careful consideration, we feel that it has merit but does not fully meet PLOS ONE’s publication criteria as it currently stands. Therefore, we invite you to submit a revised version of the manuscript that addresses the points raised during the review process.

We look forward to receiving your revised manuscript.

Kind regards,

Jun Yang

Academic Editor

PLOS ONE

Journal Requirements:

2. Regarding Data-sharing policy, it is unclear why authors have selected 'No - some restrictions will apply'.

All PLOS journals require that the minimal data set be made fully available. For more information about our data policy, please see http://journals.plos.org/plosone/s/data-availability

'The authors received no specific funding for this work.'

We note that one or more of the authors are employed by a commercial company: Hangzhou Xuelian Land Planning Co,. LTD

4. We note that Figures 3, 4 and 7-10 in your submission contain map images which may be copyrighted.

a. You may seek permission from the original copyright holder of Figures 3, 4 and 7-10 to publish the content specifically under the CC BY 4.0 license. 

6. Please include a copy of Table 3 which you refer to in your text on page 8.

Additional Editor Comments:

Reviewer 1

This paper proposes a framework for a layout evaluation of urban public sports facilities. The methods are sound, and analysis is comprehensive. However, there are still some problems:

1. Title. It is better to add your research area (Hangzhou).

2. Abstract. Condense your methods and work. Put more efforts on your research findings.

3. Introduction. “As such, there is a need to develop an approach to evaluate the spatial distribution of urban public sports facilities from the spatial equity perspective.” The author brought up with the gap, however, it lacked related literature review to prove that empirical evaluation of sports facilities distribution from spatial equity perspective is rare. Introduction needs to be enriched.

4. “2.3 Data sources”. Data (of 2017) need to be updated.

5. Gini coefficient and Lorentz curve can both describe equity. Why bother use these two indicaters, please give explanations.

6. Figures. The fonts need to be unified.

7. There are some grammar and expression errors, please polish the language in order to better convey your ideas.

Reviewer 2

This article provides a framework for the layout of urban public sports facilities. Concentric rings are created for measuring the service scopes at different levels. The Gini coefficient, lorenz curve and location entropy are employed to measure the equity of the distribution among spatial units and the intradistrict disparity. Nevertheless, there are many problems in this paper. The suggestions are given as below:

(1)I think that noticeable defects still exist in terms of innovation and its engineering application prospect. First of all, the paper about the introduction has no explanation for the current situation of measuring spatial equity, and there is no elaboration on the equity of sports facilities or the research progress of other public facilities in regions and cities. It also makes the research on the method lack of theoretical basis. The review of the current research emphasizes that the fairness of the spatial distribution of public sports facilities which mainly lies in solving the demand of people, the correlation between the differences in urban public facilities and the relationship between the scale and quantity of public sports facilities. However, the theory is more unconvincing especially in the following 2.2.2 about the Gini coefficient and Lorentz curve proposed. The novelty of the paper is difficult to be put forward on the basis of literature review without comparing with other researchers' previous work, that is, the gaps and novelty of the paper should be clearly emphasized.

(2) Figure 1, Calculation diagram of effective service area of public sports facilities. The diagram between the three should not use the same color as far as possible, because the size is relative.

(3) In part 2.2.3 of this article, the labeling of the location entropy formula is different from the description in this paper.

(4) What is the basis for " Usually, when choosing a higher-level facility, residents tend to accept higher time and distance thresholds." on page 7? " defining concentric rings for medical facilities " does not indicate the literature basis.

(5) Where is "Table 3"?

(6) Figure 3 and figure 4 are only superposition relations, which have no significance to explain separately. It is suggested that only figure 4 be used to illustrate the problem.

(7) The traditional research review in part 4.1 should be included in the introduction.

(8) This paper uses the cross-sectional data of population in 2017. The reliability and rigor of the data have yet to be agreed.

(9) The paper draws five conclusions about the impact on the layout of public sports facilities in Hangzhou through the five methods used in the article. Are these five parts overlapping? It needs to be further summarized, which is not mentioned in the conclusion.

Reviewers' comments:

Reviewer's Responses to Questions

**Comments to the Author**

1. Is the manuscript technically sound, and do the data support the conclusions?

Reviewer #1: Yes

Reviewer #2: Partly

2. Has the statistical analysis been performed appropriately and rigorously? 

Reviewer #1: Yes

Reviewer #2: Yes

3. Have the authors made all data underlying the findings in their manuscript fully available?

Reviewer #1: Yes

Reviewer #2: Yes

4. Is the manuscript presented in an intelligible fashion and written in standard English?

Reviewer #1: Yes

Reviewer #2: Yes

5. Review Comments to the Author

Reviewer #1: This paper proposes a framework for a layout evaluation of urban public sports facilities. The methods are sound, and analysis is comprehensive. However, there are still some problems:

1. Title. It is better to add your research area (Hangzhou).

2. Abstract. Condense your methods and work. Put more efforts on your research findings.

3. Introduction. “As such, there is a need to develop an approach to evaluate the spatial distribution of urban public sports facilities from the spatial equity perspective.” The author brought up with the gap, however, it lacked related literature review to prove that empirical evaluation of sports facilities distribution from spatial equity perspective is rare. Introduction needs to be enriched.

4. “2.3 Data sources”. Data (of 2017) need to be updated.

5. Gini coefficient and Lorentz curve can both describe equity. Why bother use these two indicaters, please give explanations.

6. Figures. The fonts need to be unified.

7. There are some grammar and expression errors, please polish the language in order to better convey your ideas.

Reviewer #2: This article provides a framework for the layout of urban public sports facilities. Concentric rings are created for measuring the service scopes at different levels. The Gini coefficient, lorenz curve and location entropy are employed to measure the equity of the distribution among spatial units and the intradistrict disparity. Nevertheless, there are many problems in this paper. The suggestions are given as below:

(1)I think that noticeable defects still exist in terms of innovation and its engineering application prospect. First of all, the paper about the introduction has no explanation for the current situation of measuring spatial equity, and there is no elaboration on the equity of sports facilities or the research progress of other public facilities in regions and cities. It also makes the research on the method lack of theoretical basis. The review of the current research emphasizes that the fairness of the spatial distribution of public sports facilities which mainly lies in solving the demand of people, the correlation between the differences in urban public facilities and the relationship between the scale and quantity of public sports facilities. However, the theory is more unconvincing especially in the following 2.2.2 about the Gini coefficient and Lorentz curve proposed. The novelty of the paper is difficult to be put forward on the basis of literature review without comparing with other researchers' previous work, that is, the gaps and novelty of the paper should be clearly emphasized.

(2) Figure 1, Calculation diagram of effective service area of public sports facilities. The diagram between the three should not use the same color as far as possible, because the size is relative.

(3) In part 2.2.3 of this article, the labeling of the location entropy formula is different from the description in this paper.

(4) What is the basis for " Usually, when choosing a higher-level facility, residents tend to accept higher time and distance thresholds." on page 7? " defining concentric rings for medical facilities " does not indicate the literature basis.

(5) Where is "Table 3"?

(6) Figure 3 and figure 4 are only superposition relations, which have no significance to explain separately. It is suggested that only figure 4 be used to illustrate the problem.

(7) The traditional research review in part 4.1 should be included in the introduction.

(8) This paper uses the cross-sectional data of population in 2017. The reliability and rigor of the data have yet to be agreed.

(9) The paper draws five conclusions about the impact on the layout of public sports facilities in Hangzhou through the five methods used in the article. Are these five parts overlapping? It needs to be further summarized, which is not mentioned in the conclusion.

6. PLOS authors have the option to publish the peer review history of their article (what does this mean?). If published, this will include your full peer review and any attached files.

Reviewer #1: No

Reviewer #2: No

---

## [Author Response · Author response to Decision Letter 0]

16 Jun 2021

Dear Editor:

We feel great thanks for your professional review work on our article. As you are concerned, there are several problems that need to be addressed. According to your nice suggestions, we have confirmed and modified the following problems. 

1) Thank you for providing the following statement in your Response to Reviewers document:

"Issues 3: We note your current data availability statement: "All relevant data are within the manuscript and its Supporting Information files." We also note your statement in your response to reviewers document: The paper data has been modified to be shareable.

Before we proceed, we’ll require some additional clarification to ensure your submission adheres to the PLOS ONE Data Availability policy (https://journals.plos.org/plosone/s/data-availability).

1) If you have upload your data to a repository, please also provide the relevant URLs, DOIs, or accession numbers for other researchers to access your data directly. For a list of recommended public repositories, please see https://journals.plos.org/plosone/s/recommended-repositories. 

Response：The data of this paper has been uploaded to the repository, and the URLs is https://figshare.com/s/c05872042227c1a5d9fc. If necessary, please click on the website to visit.

2) Please clarify whether you intend to make your data available publicly upon acceptance for publication. Once we receive this information, we will update your data availability statement on your behalf.

Response：Data for this paper will not be made available publicly upon acceptance for publication."

Please note that PLOS only permits data to be restricted from public access if there are ethical or legal restrictions on sharing the data. Before we proceed, we’ll require some additional information to ensure your submission adheres to the PLOS policy on acceptable data access restrictions: https://journals.plos.org/plosone/s/data-availability#loc-acceptable-data-access-restrictions.

1) Please confirm whether there are legal or ethical restrictions on sharing your data publicly.

2) If legal or ethical restrictions apply, please provide all necessary instructions and non-author contact information (preferably email) for a data access committee, ethics committee, or other institutional body other researchers would require to apply for data access. Note that it is not acceptable for an author to be the sole named individual responsible for ensuring data access.

3) If there are no legal or ethical restrictions on sharing your data publicly, please upload the minimal anonymized data set necessary to replicate your study findings as either Supporting Information files, or to a stable public repository. If you upload your data to a repository, please also provide the relevant URLs, DOIs, or accession numbers for other researchers to access your data directly. For a list of recommended public repositories, please see https://journals.plos.org/plosone/s/recommended-repositories.

4) We note that the DOI provided (10.6084/m9.figshare.14754552) does note direct to your data set(s). If there are no legal or ethical restrictions on sharing your data publicly, please ensure that any URLs, DOIs, or accession numbers you provide direct to your data.

Response：There are no legal or ethical restrictions on sharing our data publicly. The URLs is https://figshare.com/s/c05872042227c1a5d9fc. We have chicked that the URLs directs to the data.

Thank you again for your positive comments and valuable suggestions to improve the quality of our manuscript !

---

## [Decision Letter · Decision Letter 1]

6 Jul 2021

PONE-D-21-10518R1

Spatial equity in the layout of urban public sports facilities in Hangzhou

PLOS ONE

Dear Dr. Ding,

Thank you for submitting your manuscript to PLOS ONE. After careful consideration, we feel that it has merit but does not fully meet PLOS ONE’s publication criteria as it currently stands. Therefore, we invite you to submit a revised version of the manuscript that addresses the points raised during the review process.

We look forward to receiving your revised manuscript.

Kind regards,

Jun Yang

Academic Editor

PLOS ONE

Additional Editor Comments (if provided):

Reviewer 1

In the "Response to Reviewers" document, I can only see your response to the editor. Please upload your "response to reviewers" so that we can see your detailed modification and reasons according to the reviewers' comments.

Also, please refer to this literature which may be helpful to you. "Study on the Impact of High-speed Railway Opening on China's Accessibility Pattern and Spatial Equality[J].Sustainability 2018,10,2943. doi:10.3390/su10082943".

Reviewer 2

The article has been greatly improved after modification. No matter from the review of research literature or the clear innovation point, the work done is worth affirming. However, there are still some small problems that the author should consider carefully. The details are as follows:

(1) What is the reference basis or literature of formula 2 and 3? Please indicate.

(2) The format of reference is chaotic, so it is necessary to unify the format, pay attention to the abbreviation mode and the requirements of symbols and spaces.

(3) The clarity of the article pictures should be adjusted uniformly.

Reviewers' comments:

Reviewer's Responses to Questions

**Comments to the Author**

1. If the authors have adequately addressed your comments raised in a previous round of review and you feel that this manuscript is now acceptable for publication, you may indicate that here to bypass the “Comments to the Author” section, enter your conflict of interest statement in the “Confidential to Editor” section, and submit your "Accept" recommendation.

Reviewer #1: All comments have been addressed

Reviewer #2: All comments have been addressed

2. Is the manuscript technically sound, and do the data support the conclusions?

Reviewer #1: Yes

Reviewer #2: Partly

3. Has the statistical analysis been performed appropriately and rigorously? 

Reviewer #1: Yes

Reviewer #2: Yes

4. Have the authors made all data underlying the findings in their manuscript fully available?

Reviewer #1: Yes

Reviewer #2: Yes

5. Is the manuscript presented in an intelligible fashion and written in standard English?

Reviewer #1: Yes

Reviewer #2: No

6. Review Comments to the Author

Reviewer #1: In the "Response to Reviewers" document, I can only see your response to the editor. Please upload your "response to reviewers" so that we can see your detailed modification and reasons according to the reviewers' comments.

Also, please refer to this literature which may be helpful to you. "Study on the Impact of High-speed Railway Opening on China's Accessibility Pattern and Spatial Equality[J].Sustainability 2018,10,2943. doi:10.3390/su10082943".

Reviewer #2: The article has been greatly improved after modification. No matter from the review of research literature or the clear innovation point, the work done is worth affirming. However, there are still some small problems that the author should consider carefully. The details are as follows:

(1) What is the reference basis or literature of formula 2 and 3? Please indicate.

(2) The format of reference is chaotic, so it is necessary to unify the format, pay attention to the abbreviation mode and the requirements of symbols and spaces.

(3) The clarity of the article pictures should be adjusted uniformly.

7. PLOS authors have the option to publish the peer review history of their article (what does this mean?). If published, this will include your full peer review and any attached files.

Reviewer #1: No

Reviewer #2: No

---

## [Author Response · Author response to Decision Letter 1]

21 Jul 2021

Response to Reviewers of the Second Review

Dear Editor:

We feel great thanks for your professional review work on our article. As you are concerned, there are several problems that need to be addressed. According to your nice suggestions, we have made further corrections to our previous draft, the detailed corrections are listed below.

Reviewer 1

Issues 1: In the "Response to Reviewers" document, I can only see your response to the editor. Please upload your "response to reviewers" so that we can see your detailed modification and reasons according to the reviewers' comments.

Response："Response to Reviewers" document of the first review is on the next page.

Issues 2: Also, please refer to this literature which may be helpful to you. "Study on the Impact of High-speed Railway Opening on China's Accessibility Pattern and Spatial Equality[J].Sustainability 2018,10,2943. doi:10.3390/su10082943".

Response：This literature has some enlightening significance on the research ideas and methods of this paper and has been cited.

Reviewer 2

The article has been greatly improved after modification. No matter from the review of research literature or the clear innovation point, the work done is worth affirming. However, there are still some small problems that the author should consider carefully. The details are as follows:

Issues 1: What is the reference basis or literature of formula 2 and 3? Please indicate.

Response： Reference basis of Formula 2 and 3 is added in Section 2.2.2 and 2.2.3 of this paper.

Issues 2: The format of reference is chaotic, so it is necessary to unify the format, pay attention to the abbreviation mode and the requirements of symbols and spaces.

Response：The format of reference has been modified according to the requirements of the journal. 

Issues 3: The clarity of the article pictures should be adjusted uniformly.

Response：The clarity of the pictures in this paper has been adjusted uniformly, and the pictures with poor clarity have been replaced. But the pictures in PDF are still not clear, so we uploaded figures in “orther” item.

 

Response to Reviewers of the First Review

Dear Editor:

We feel great thanks for your professional review work on our article. As you are concerned, there are several problems that need to be addressed. According to your nice suggestions, we have made extensive corrections to our previous draft, the detailed corrections are listed below.

Response to Journal Requirements:

Requirement 1：Please ensure that your manuscript meets PLOS ONE's style requirements, including those for file naming. The PLOS ONE style templates can be found at

Response：The paper has been modified according to PLoS One's Style Requirements.

Requirement 2：Regarding Data-sharing policy, it is unclear why authors have selected 'No - some restrictions will apply'.

Response：The paper data has been modified to be shareable.

Requirement 3：Thank you for stating the following in the Financial Disclosure section.

Response：It has been explained in the cover letter that the author once studied as a graduate student in our school and volunteered to participate in the research after graduation. And check the contribution of the author in the online submission form, which is consistent with the voluntary statement. We compilated relevant contents in the Competing Interests Statement.

Requirement 4：We note that Figures 3, 4 and 7-10 in your submission contain map images which may be copyrighted.

Response： Maps are public data of the government. The source of the base map has been added in the revised manuscript, including the name of the website and the website address, which explains that it is public and does not involve copyright.

Requirement 5：PLOS requires an ORCID iD for the corresponding author in Editorial Manager on papers submitted after December 6th, 2016. Please ensure that you have an ORCID iD and that it is validated in Editorial Manager. To do this, go to ‘Update my Information’ (in the upper left-hand corner of the main menu), and click on the Fetch/Validate link next to the ORCID field. This will take you to the ORCID site and allow you to create a new iD or authenticate a pre-existing iD in Editorial Manager. Please see the following video for instructions on linking an ORCID iD to your Editorial Manager account: https://www.youtube.com/watch?v=_xcclfuvtxQ

Response：The ORCID ID has been set.

6. Please include a copy of Table 3 which you refer to in your text on page 8.

Response： Table 3 has been supplemented

Response to Reviewers 1

This paper proposes a framework for a layout evaluation of urban public sports facilities. The methods are sound, and analysis is comprehensive. However, there are still some problems:

Comment 1： Title. It is better to add your research area (Hangzhou).

Response：The research area of this paper (Hangzhou) has been added in the title.

Comment 2： Abstract. Condense your methods and work. Put more efforts on your research findings.

Response：As can be seen in the abstract of the revised manuscript, this paper has simplified the statement of the research methods and put emphasis on the elaboration of the results.

Comment 3： Introduction. “As such, there is a need to develop an approach to evaluate the spatial distribution of urban public sports facilities from the spatial equity perspective.” The author brought up with the gap, however, it lacked related literature review to prove that empirical evaluation of sports facilities distribution from spatial equity perspective is rare. Introduction needs to be enriched.

Response：This paper has re-modified the caontent of literature review in the introduction part. After literature review, There are few cases of the empirical evaluation of the distribution of sports facilities from the perspective of spatial equity. This paper finds that the spatial equity research of sports facilities mainly reflects in three aspects: Equalization, accessibility and optimized layout of facilities.

However, the research on sports facilities is almost blank in the field that involved the “space matching” between facilities and residents, and the research methods are usually qualitative research, which has a certain hysteresis quality. In other words, the traditional urban public sports facilities planning adopts per capita index to try to ensure that the spatial allocation of public facilities reaches the goal of social equity, but lacks an effective method to evaluate the "spatial matching" of facilities and resident population distribution. In view of this, this paper proposed research design.

Comment 4：“2.3 Data sources”. Data (of 2017) need to be updated.

Response：China's population census is conducted every ten years. At present, the sixth census(2010) data is too old and the seventh census(2020) data has not yet been released. In 2017, Hangzhou municipal government conducted a miniature population By-census. The data is made public by the government(https://data.hz.zjzwfw.gov.cn/) and is the latest data before the 7th census is released. Hangzhou is a city with a population of ten million. A census of a large city with a population of nearly ten million is not conducted every year. Therefore, Based on the minimum research unit in this paper, the demographic data in 2017 are the most recent available.

Meanwhile, urban public sports facilities, while having access to the latest data, are still used for the data of 2017 in order to be consistent with population data. The main purpose of this paper is to establish an a evaluation framework. So although there is a gap of 3 years between the data used in 2017 and the reality, the data in this paper should still be applicable to the method construction.

Comment 4：Gini coefficient and Lorentz curve can both describe equity. Why bother use these two indicaters, please give explanations.

Response：Both the Gini coefficient and the Lorentz curve are used to measure the degree of inequality of distribution, but they have different forms of expression.Gini coefficient is an overall numerical indicator, and is more concise. However，it does not reflect superfluous information. Lorenz curve shows the distribution of public sports facilities in the permanent resident population in a graphical way, and it can investigate the proportion of the permanent resident population enjoying the resources of public sports facilities. So it's an extension of the interpretation of Gini coefficient. The above description has been added to 2.2.2.

Comment 6：Figures. The fonts need to be unified.

Response：Figures and fonts have been unified in accordance with the format requirements.

Comment 7：There are some grammar and expression errors, please polish the language in order to better convey your ideas.

Response：This article has been polished to make it easier for readers to understand.

Response to Reviewers 2

This article provides a framework for the layout of urban public sports facilities. Concentric rings are created for measuring the service scopes at different levels. The Gini coefficient, lorenz curve and location entropy are employed to measure the equity of the distribution among spatial units and the intradistrict disparity. Nevertheless, there are many problems in this paper. The suggestions are given as below:

Comment 1： I think that noticeable defects still exist in terms of innovation and its engineering application prospect. First of all, the paper about the introduction has no explanation for the current situation of measuring spatial equity, and there is no elaboration on the equity of sports facilities or the research progress of other public facilities in regions and cities. It also makes the research on the method lack of theoretical basis. The review of the current research emphasizes that the fairness of the spatial distribution of public sports facilities which mainly lies in solving the demand of people, the correlation between the differences in urban public facilities and the relationship between the scale and quantity of public sports facilities. However, the theory is more unconvincing especially in the following 2.2.2 about the Gini coefficient and Lorentz curve proposed. The novelty of the paper is difficult to be put forward on the basis of literature review without comparing with other researchers' previous work, that is, the gaps and novelty of the paper should be clearly emphasized.

Response：This paper has re-modified the caontent of literature review in the introduction part，

and has reviewed the research on spatial equity of public services again. And what we found was that the research was focused on three aspects: equalization, accessibility and optimized layout of facilities. The subjects included green space, parks, medical facilities, rail transit, and more. At the same time, many quantitative research methods, such as the Gini coefficient and the Lorentz curve, are used. Specific content has been elaborated in the introduction part.

Compared with public service facilities, there are relatively few studies on the spatial equity of public sports facilities, especially when it comes to the "spatial matching" between facilities and residents. The research methods are mostly qualitative studies with a certain lag. In other words, the traditional urban public sports facilities planning adopts per capita index to try to ensure that the spatial allocation of public facilities reaches the goal of social equity, but lacks an effective method to evaluate the "spatial matching" of facilities and resident population distribution. In view of this, this paper is based on ArcGIS analysis platform, and using the methods of gini coefficient, lorenz curve and location entropy, trying to construct evaluation system to measure the spatial equity of urban public sports facilities, so that the results can be quantified, visualization, and systemically and universality, in order to provide some reference for the layout planning of public sports facilities in different cities.

Comment 2： Figure 1, Calculation diagram of effective service area of public sports facilities. The diagram between the three should not use the same color as far as possible, because the size is relative.

Response： Three colors have been used to represent the effective service area of three levels of public sports facilities.

Comment 3：In part 2.2.3 of this article, the labeling of the location entropy formula is different from the description in this paper.

Response：The interpretation of the location entropy formula in 2.2.3 has been modified.

Comment 4： What is the basis for " Usually, when choosing a higher-level facility, residents tend to accept higher time and distance thresholds." on page 7? " defining concentric rings for medical facilities " does not indicate the literature basis.

Response：Compared with the lower-level sports facilities, the sports services and functions provided by the higher-level sports facilities are more abundant and complete. According to this, residents have a stronger and specific purpose for travel, and the travel modes will be more diversified. Therefore, when choosing a higher-level facility, residents tend to accept higher time and distance thresholds.

To the question of the literature basis of " defining concentric rings for medical facilities ", the corresponding literature is added to explain it. Through the definition of service radius of sports facilities in national standards and the description of travel characteristics in relevant literature, we comprehensively determined our time and distance corresponding to different modes for moving.

Comment 5： Where is "Table 3"?

Response：Due to a personal oversight, “Table 3” was accidentally deleted during formatting and has been added.

Comment 6： Figure 3 and figure 4 are only superposition relations, which have no significance to explain separately. It is suggested that only figure 4 be used to illustrate the problem.

Response：We accepted your pertinent suggestions. In order to demonstrate the spatial characteristics of service level of sports facilities at all levels separately, Figure 3 was incorporated into Figure 4 for unified expression, and the content of Figure 4 was emphasized in the figure. 

Comment 7：The traditional research review in part 4.1 should be included in the introduction.

Response：This paper has re-modified the caontent of literature review in the introduction part.

This paper discusses the methods used in the study of space equity between public sports facilities and other public service facilities. It is found that there is a big gap in public sports facilities and there are relatively few empirical studies using quantitative methods.

Comment 8：This paper uses the cross-sectional data of population in 2017. The reliability and rigor of the data have yet to be agreed.

Response：The cross-sectional data of population in 2017 are from the publicly available government data and open data of Internet platform. China's population census is conducted every ten years. In 2017, between the 6th(2010) and 7th(2020) censuses, Hangzhou municipal government conducted a miniature population By-census. The data is made public by the government(https://data.hz.zjzwfw.gov.cn/) and is the latest data before the 7th census is released. However, the public data of the government in 2017 were only counted at the street level. In order to obtain the population data at the community level, this paper needed to rely on Baidu Map Open Platform, and crawl Baidu heat map of the population at night of each season in 2017 for representing the permanent population living in the local. The network data is imported into ArcGIS, and the raster calculator is used for overlay and average calculation to get the final layer data. Finally, combined with the existing population data at the street level, the population of the community unit is determined according to the proportion of the area of heat grid in the community unit to the total area of the street in which the community is located. 

The main purpose of this study is to build a research framework, so in a way，this method is applicable to measure the scale of community population. 

Comment 9：The paper draws five conclusions about the impact on the layout of public sports facilities in Hangzhou through the five methods used in the article. Are these five parts overlapping? It needs to be further summarized, which is not mentioned in the conclusion.

Response： According to the research design and the research methods used in this paper, the conclusions are modified, elaborated in detail from three levels, and summarized. 

This paper constructs a complete research framework for evaluating the spatial equity of facilities, which is mainly divided into three levels. Firstly, the quantitative index of sports facilities resource level is obtained by using buffer analysis and superposition analysis. Secondly, Gini coefficient and Lorenz curve are used to reflect the equity of public sports facilities resources in the spatial distribution of the whole resident population. Finally, in order to present the results more finely, the spatial pattern of equity of public sports facilities was analyzed by using location entropy method. 

The above three levels are layers of progressive relationship, and there is no overlap

Thank you again for your positive comments and valuable suggestions to improve the quality of our manuscript !

---

## [Decision Letter · Decision Letter 2]

2 Aug 2021

Spatial equity in the layout of urban public sports facilities in Hangzhou

PONE-D-21-10518R2

Dear Dr. Ding,

We’re pleased to inform you that your manuscript has been judged scientifically suitable for publication and will be formally accepted for publication once it meets all outstanding technical requirements.

Kind regards,

Jun Yang

Academic Editor

PLOS ONE

Additional Editor Comments (optional):

Accept

Reviewers' comments:

Reviewer's Responses to Questions

**Comments to the Author**

1. If the authors have adequately addressed your comments raised in a previous round of review and you feel that this manuscript is now acceptable for publication, you may indicate that here to bypass the “Comments to the Author” section, enter your conflict of interest statement in the “Confidential to Editor” section, and submit your "Accept" recommendation.

Reviewer #1: All comments have been addressed

Reviewer #2: (No Response)

2. Is the manuscript technically sound, and do the data support the conclusions?

Reviewer #1: Yes

Reviewer #2: (No Response)

3. Has the statistical analysis been performed appropriately and rigorously? 

Reviewer #1: Yes

Reviewer #2: (No Response)

4. Have the authors made all data underlying the findings in their manuscript fully available?

Reviewer #1: Yes

Reviewer #2: (No Response)

5. Is the manuscript presented in an intelligible fashion and written in standard English?

Reviewer #1: Yes

Reviewer #2: (No Response)

6. Review Comments to the Author

Reviewer #1: (No Response)

Reviewer #2: (No Response)

7. PLOS authors have the option to publish the peer review history of their article (what does this mean?). If published, this will include your full peer review and any attached files.

Reviewer #1: No

Reviewer #2: No

---

## [Editor Report · Acceptance letter]

9 Aug 2021

PONE-D-21-10518R2 

Spatial equity in the layout of urban public sports facilities in Hangzhou 

Dear Dr. Ding:

I'm pleased to inform you that your manuscript has been deemed suitable for publication in PLOS ONE. Congratulations! Your manuscript is now with our production department. 

Kind regards, 

on behalf of

Dr. Jun Yang 

Academic Editor

PLOS ONE